# Rate-Distortion Theoretic Bounds on Generalization Error for Distributed Learning

**Milad Sefidgaran** [†], **Romain Chor** [†], **Abdellatif Zaidi** [†][‡]

[†] Paris Research Center, Huawei Technologies France

[‡] Université Paris-Est, Champs-sur-Marne 77454, France

{milad.sefidgaran2, romain.chor}@huawei.com, abdellatif.zaidi@u-pem.fr

## Abstract

In this paper, we use tools from rate-distortion theory to establish new upper bounds on the generalization error of statistical distributed learning algorithms. Specifically, there are $K$ clients whose individually chosen models are aggregated by a central server. The bounds depend on the compressibility of each client's algorithm while keeping other clients' algorithms un-compressed, and leveraging the fact that small changes in each local model change the aggregated model by a factor of only $1/K$. Adopting a recently proposed approach by Sefidgaran et al., and extending it suitably to the distributed setting, enables smaller rate-distortion terms which are shown to translate into tighter generalization bounds. The bounds are then applied to the *distributed* support vector machines (SVM), *suggesting* that the generalization error of the distributed setting decays faster than that of the centralized one with a factor of $\mathcal{O}(\sqrt{\log(K)/K})$. This finding is validated also experimentally. A similar conclusion is obtained for a *multiple-round federated learning* setup where each client uses *stochastic gradient Langevin dynamics* (SGLD).

## 1 Introduction

A key performance indicator of any stochastic learning algorithm that uses a given finite set of data points is how well it performs on points that are outside that set, i.e., unseen data. This is often captured through the so-called *generalization error*. The questions of what really controls the generalization error of a given stochastic algorithm, and how to make it sufficiently small, are still not yet well understood, however. For example, while classic approaches [SSBD14] suggest that algorithms with over-parameterized models are likely to *overfit*, it is now known that there exist a few such ones which do generalize well [ZBH+17]. Common approaches to studying the generalization error of a statistical learning algorithm often consider the *effective hypothesis space* induced by the algorithm, rather than the entire hypothesis space, or the information leakage about the *training dataset*. Examples include information-theoretic (mutual information) approaches [RZ16, XR17, HRVSG21, HDMR21, NHD+20, SZ20], compression-based approaches [AGNZ18, SAN20, HJTW21, BSE+21, KLG+21] and intrinsic-dimension or fractal based approaches [ŞSDE20, BLGŞ21, HŞKM21]. Recently, a novel approach [SGRS22] that generalizes the notion of algorithm compressibility by using lossy covering from source coding concepts was used to show that the compression error rate of an algorithm is strongly connected to its generalization error both in-expectation and with high probability; and, consequently, establish new rate-distortion-based bounds on the generalization error. The bounds of [SGRS22] were shown to possibly improve strictly upon those of [XR17, BZV20] and [SZ20]. The approach also has the advantage to offer a unifying perspective on mutual information, compressibility, and fractal-based frameworks.

Another major focus of machine learning research over recent years has been the study of statistical learning algorithms when applied in distributed (network or graph) settings. In part, this is due to the

emergence of new applications in which resources are constrained, data is distributed or the need to preserve privacy [VWK+20, MMR+17, KMY+16, KLN+11]. Reducing the computational complexity by offloading the model training using algorithms such as *parallel* stochastic gradient descent is another reason for the popularity of these algorithms. Generally, that results in extra communication costs, however [ZZ15, AGL+17, CCB+18, LM22, WLC22]. Example such algorithms include the now popular *Federated Learning* [YAE+18, KMA+19, LSTS20, RCZ+20, KKM+20, YZR21], the *Split Learning* of [GR18], *Group Alternating Direction Method of Multipliers* [EPB+20], and the so-called *in-network learning* of [AZ19, MZ21]. Despite its importance, however, little is known about the generalization guarantees of distributed statistical learning algorithms. In fact, in the distributed learning setting the technical challenges are numerous, including the lack of a proper definition of the generalization error in this case [YMNS22, MSS19]. Notable exception works in this direction include [YDVP20] and [BDP22]. In [YDVP20], information-theoretic upper bounds on the generalization error of distributed statistical learning algorithms are obtained merely by viewing the entire distributed system, from the input data to each (local) algorithm to the output aggregated model, as a single (centralized) algorithm and applying to it the bounds of [XR17, BZV20]. While somewhat useful this, however, has left the difference between the bounds for the distributed learning setting and their counterparts for the centralized learning setting implicit in the involved mutual information terms. The problem of studying the generalization error of distributed statistical learning algorithms was further studied in the recent [BDP22]. Therein, using results from [BZV20], the authors establish bounds on the expectation of the generalization error for two special cases. For the first, linear or location models with Bregman divergence loss, the proposed bound on the generalization error for the distributed setup [BDP22, Theorem 4] is shown better (i.e., smaller) than its counter-part for the centralized learning setting by a factor of $\mathcal{O}(1/\sqrt{K})$, where $K$ is the number of clients. This result, however, relies strongly on the assumed linearity of the loss with respect to the hypothesis. For the second, Lipschitz continuous loss [BDP22, Theorem 5], similar behavior is shown by reducing the problem to the centralized case and using the triangle inequality.

In this work, we study the generalization error of distributed statistical learning algorithms. Essentially we extend suitably the approach of [SGRS22] to establish rate-distortion theoretic upper bounds on the generalization error. In doing so, we bring the analysis of the distributed architecture of the learning problem into the bounds. The bounds, which hold with high probability and in-expectation, allow to only consider the compressibility of each local algorithm – the latter having an effect with a factor of only $1/K$ on the aggregated model order-wise. This is shown to result in a more relaxed distortion criterion for the *local algorithm compressibility*, smaller rate-distortion terms; and, in turn, better generalization bounds. Furthermore, we apply our results to the *distributed support vector machines* (DSVM). The obtained bounds suggest that for the non-separable data, the generalization error of the distributed setting decays faster than that of the centralized one with a factor of $\sqrt{\log(K)/K}$. We conducted experiments on DSVM that confirm this finding. We also consider the related Federated learning setting, and derive bounds on the generalization error in two setups: when each client applies the *stochastic gradient Langevin dynamics* (SGLD) method and locally deterministic algorithms with Lipschitz loss (in Appendix C.1). In all cases, our bounds suggest a decreasing behavior for the generalization performance as $K$ grows.

**Notation.** Random variables, their realizations, and their domains are denoted respectively by upper-case, lower-case, and calligraphic fonts, *e.g.,* $X$, $x$, and $\mathcal{X}$. Their distributions and expectations are denoted by $P_X$ and $\mathbb{E}[X]$. The random variable $X$ is called $\sigma$-*subgaussian* if for all $t \in \mathbb{R}$, $\mathbb{E}[\exp(t(X - \mathbb{E}[X]))] \leqslant \exp(\sigma^2 t^2/2)$, *e.g.,* if $X \in [a, b]$, then $X$ is $\frac{b-a}{2}$-subgaussian. A vector of $m \in \mathbb{N}$ numbers (or random variables) $(x_1, \ldots, x_m)$ are denoted by either $x_{1:m}$ or $\{x_i\}_{i=1}^m$, depending on the context, and the vector $(x_1, \ldots, x_{i-1}, x_{i+1}, \ldots, x_m)$ is denoted by $x_{1:m \backslash i}$. Similarly for $n, m \in \mathbb{N}$, a vector $(x_{1,1}, \ldots, x_{1,m}, x_{2,1}, \ldots, x_{2,m}, x_{n,1}, \ldots, x_{n,m})$ is denoted by $x_{1:n,1:m}$ or $\{x_{i,1:m}\}_{i \in [n]}$, or $\{x_{1:n,j}\}_{j \in [m]}$, where $[n] = \{1, \ldots, n\}$. Parts of our results are stated in terms of information-theoretic quantities: for random variables $X$ and $Y$, we denote the differential *entropy* of $X$ by $h(X)$, the *conditional differential entropy* of $X$ given $Y$ by $h(X|Y)$, and the mutual information between them by $I(X; Y)$. Moreover, the *Kullback–Leibler* (KL) divergence between distributions $Q$ and $P$ is denoted by $D_{KL}(Q \| P)$. For more details, we refer the reader to [CT06, PW14].

## 2 Preliminaries and problem setup

For convenience, we start with a brief review of the standard (centralized) statistical learning setup together with a few definitions and recent results associated with it. Let the *input data* $Z$ be

distributed according to an unknown distribution $\mu$ over the *data space* $\mathcal{Z}$. A *training dataset* $S = (Z_1, \ldots, Z_n) \sim \mu^{\otimes n}$ consists of $n$ samples $\{Z_i\}$ generated independently each according to $\mu$. A possibly stochastic learning algorithm $\mathcal{A} \colon \mathcal{Z}^n \mapsto \mathcal{W}$ (e.g., stochastic gradient descent) assigns an hypothesis $\mathcal{A}(S) = W$ chosen from the *hypothesis class* $\mathcal{W} \subseteq \mathbb{R}^d$ to every $S \in \mathcal{Z}^n$. The map $\mathcal{A}$ induces a conditional distribution $P_{W|S}$ which together with $\mu$ induce the joint dataset-hypothesis distribution $P_{S,W} = \mu^{\otimes n} P_{W|S}$. The quality of the prediction is measured using a loss function $\ell \colon \mathcal{Z} \times \mathcal{W} \mapsto \mathbb{R}^+$. The *generalization error* of an algorithm $\mathcal{A}$ is defined as $\mathrm{gen}(s, w) \coloneqq \mathcal{L}(w) - \hat{\mathcal{L}}(s, w)$ where $\mathcal{L}(w) \coloneqq \mathbb{E}_{Z \sim \mu}[\ell(z, w)]$ denotes the population risk and $\hat{\mathcal{L}}(s, w) \coloneqq \frac{1}{n} \sum_{j \in [n]} \ell(z_j, w)$ denotes the empirical risk. Note that the generalization error depends on the loss function $\ell(z, w)$, underlying distribution $\mu$, the sample size $n$, and also the learning algorithm $P_{W|S}$. In the binary classification context, where $\mathcal{Z} = \mathcal{X} \times \mathcal{Y}$ and $\mathcal{Y} = \{-1, +1\}$, we often consider the 0-1 loss function $\ell_0(z, w) \coloneqq \mathbb{1}_{\{yf(x,w)<0\}}$, where the sign of $f(x, w)$, $f \colon \mathcal{X} \times \mathcal{W} \mapsto \mathbb{R}$, is the label prediction by hypothesis $w$ and $\mathbb{1}$ is the indicator function. In this setup, it is common to assess the empirical risk with respect to 0-1 loss function with margin $\theta \in \mathbb{R}^+$ defined as $\ell_\theta(z, w) \coloneqq \mathbb{1}_{\{yf(x,w)<\theta\}}$, while using 0-1 loss function for the population risk evaluation. We denote the corresponding empirical risk as $\hat{\mathcal{L}}_\theta(s, w) \coloneqq \frac{1}{n} \sum_{j \in [n]} \ell_\theta(z_j, w)$ and the generalization error as $\mathrm{gen}_\theta(s, w) \coloneqq \mathcal{L}(w) - \hat{\mathcal{L}}_\theta(s, w)$.

The exact analysis of the generalization error $\mathrm{gen}(S, W)$ seems out of reach; and, for this reason, as already mentioned upper bounds on it were developed using an information-theoretic approach. Essentially, such an approach connects the generalization error of a statistical learning algorithm $\mathcal{A}$ with the mutual information between the input data sample $S$ and the algorithm output $W = \mathcal{A}(S)$. For details, the reader may refer to a line of work that was initiated by Russo and Zou [RZ16] and Xu and Raginsky [XR17] and since then improved by using various conditional versions of mutual information. Other approaches rely on the observation that the output $W$ can be *compressible* in some suitable sense [AGNZ18] or the algorithm might generate a *fractal structure* [ŞSDE20].

Very recently, an approach [SGRS22] that relies on probabilistic $\epsilon$-covering from source coding concepts was proposed and shown to possibly improve strictly over the aforementioned, seemingly unrelated, approaches, while offering a unifying framework to them. The upper bounds of [SGRS22] are rate-distortion theoretic. Specifically, let $\hat{\mathcal{W}}$ be the alphabet of the *compressed hypothesis* and $\hat{\ell} \colon \mathcal{Z} \times \hat{\mathcal{W}} \mapsto \mathbb{R}^+$ a loss function (possibly different from $\ell$). Accordingly, for $\hat{w} \in \hat{\mathcal{W}}$ and $s \in \mathcal{Z}^n$, let $\mathrm{gen}(s, \hat{w})$ be defined with respect to $\hat{\ell}$. For every distribution $Q$ defined over $\mathcal{S} \times \mathcal{W}$, the rate-distortion function with respect to $\hat{\mathcal{W}}$ is defined as

$$\mathfrak{RD}(Q, \epsilon) \coloneqq \inf_{P_{\hat{W}|S}} I(S; \hat{W}),$$

$$\text{s.t.} \quad \mathbb{E}_{(S,W) \sim Q}[\mathrm{gen}(S, W)] - \mathbb{E}_{(S,\hat{W}) \sim Q_S P_{\hat{W}|S}}[\mathrm{gen}(S, \hat{W})] \leqslant \epsilon. \tag{1}$$

where $Q_S$ is the $Q$-marginal of $S$, and the infimum is taken over all Markov kernels (conditional distributions) of a random variable $\hat{W} \in \hat{\mathcal{W}}$ given $S$. Note that $P_{S,W}$ induced by the algorithm $\mathcal{A}$ is a particular case of $Q$. In the following, we shortly discuss the related intuition and concept behind the above terms. The reader is referred to Appendix A for more details on this.

This rate-distortion function $\mathfrak{RD}(Q, \epsilon)$ is the adaptation of the rate-distortion function emerged in the lossy source compression context [Ber75, CT06] to stochastic learning algorithms [SGRS22]. In the lossy source compression context [Ber75, CT06], this function quantifies the fundamental compression rate of a source $X \sim P_X$ to within some desired average distortion level $\epsilon$. To this end, infinitely many i.i.d. instances of the source ($\{X_i\}_{i \in [m]}$, $X_i \sim P_X$, and $m \to \infty$) are compressed (quantized) simultaneously. The joint compression approach is known as the *block-coding* technique. In the learning context, this term quantifies the *lossy algorithm compressibility* [SGRS22], with the following intuition for the case where $\mathcal{Z}$ and $\mathcal{W}$ are finite sets. Each $P_{\hat{W}|S}$ denotes a learning algorithm and the distortion criterion $\mathbb{E}[\mathrm{gen}(S, W) - \mathrm{gen}(S, \hat{W})] \leqslant \epsilon$ guarantees that in average the difference between the generalization error of the original algorithm $P_{W|S}$ and that of the "compressed" algorithm $P_{\hat{W}|S}$ does not exceed $\epsilon$. The rate-distortion term $\mathfrak{RD}(P_{S,W}, \epsilon)$ quantifies the lossy compressibility of the algorithm $P_{W|S}$ in the following sense: for large $m \in \mathbb{N}$ and every admissible $P_{\hat{W}|S}$, a compressed hypothesis space of size $N_m \approx e^{mI(S;\hat{W})}$ can be found such that, with high probability, for each $m$ i.i.d. instances of the original algorithm $P_{W|S}$ there exists at least one compressed hypothesis for which the difference between the average generalization error over

the $m$ i.i.d. instances of $P_{W|S}$ at hand and that of the found hypothesis does not exceed $\epsilon$. Recall the tail bound on the generalization error of [SGRS22, Theorem 10].

**Theorem 1** ([SGRS22, Theorem 10] )**.** *Suppose that the learning algorithm $\mathcal{A}(S)$ induces $P_{S,W}$ and suppose that for all $\hat{w} \in \hat{\mathcal{W}}$, $\hat{\ell}(Z, \hat{w})$ is $\sigma$-subgaussian. Then, for any fixed $\epsilon \in \mathbb{R}$ and $\delta \geqslant 0$, with probability at least $1 - \delta$, $\text{gen}(S, W) \leqslant \sqrt{2\sigma^2 (R_p(\delta, \epsilon) + \log(1/\delta))/n} + \epsilon$, where $R_p(\delta, \epsilon) :=$ $\sup_{Q : \, D_{KL}(Q\|P_{S,W}) \leqslant \log(1/\delta)} \mathfrak{RD}(Q, \epsilon)$ and the supremum is over all possible distributions over $\mathcal{S} \times \mathcal{W}$.*

The above bound depends not only $\mathfrak{RD}(P_{S,W}, \epsilon)$ but also on $\mathfrak{RD}(Q, \epsilon)$ terms for all distributions $Q$ that are close enough to $P$. In other words, to guarantee a good generalization performance, the compressibility of the algorithm under every such small perturbation of $P_{S,W}$ needs to be considered.

**Remark 1.** *The reader may notice two (minor) differences between Theorem 1 as stated here and [SGRS22, Theorem 10]. First, we here express the rate-distortion terms with respect to the distortion function $d(w, \hat{w}; s) := \text{gen}(s, w) - \text{gen}(s, \hat{w})$ instead of a possibly smaller distortion function $d'(w, \hat{w}; s) := \inf_{(s', w') \in \text{supp}(Q)} \text{gen}(s', w') - \text{gen}(s, \hat{w})$ that was considered in [SGRS22], where $\text{supp}(Q)$ denotes the support of $Q$. Even though the latter could possibly lead to stronger results, we do not consider it here because it is less amenable to computations. Second, in Theorem 1 the loss function $\hat{\ell}(z, \hat{\tilde{w}})$ is allowed to differ from the original $\ell(z, w)$; and this possibly leads to some (rather small) improvement of the result of [SGRS22, Theorem 10]. The mentioned two small differences, however, do not require any change in the proof of [SGRS22, Theorem 10].*

**Remark 2.** *Similar to [SGRS22, Theorem 10], most of our results that will follow in this paper require the subgaussianity assumption of the loss function to hold. In both cases the assumption is used to properly bound the moment generating function (MGF) of a specific zero-mean random variable using the Hoeffding inequality (see [SGRS22, Section E.6.2] for the details) . Alternatively, this MGF can be upper bounded using approaches similar to [BZV20].*

**Problem setup.** In this work, we consider a *homogeneous distributed learning* setup that exploits the participation of $K$ clients, as described in the following. Each client $i \in [K]$ has access to a training dataset $S_i = \{Z_{i,1}, \ldots, Z_{i,n}\} \sim \mu^{\otimes n}$ of size $n$, drawn independently of each other and independently of other clients' training datasets from the same distribution $\mu$. The local learning algorithm $\mathcal{A}_i$ at each client picks a hypothesis $\mathcal{A}_i(S_i) = W_i \in \mathcal{W}_i = \mathcal{W}$ according to $P_{W_i|S_i}$. The induced joint distribution of $(S_i, W_i)$ is denoted by $P_{S_i, W_i}$. The server receives the hypotheses $W_{1:K}$ and picks the hypothesis $\overline{W}$ as

$$\overline{W} := (W_1 + \cdots + W_K)/K.$$

We denote the distributed learning algorithm as $\mathcal{A}_{1:K}(S_{1:K})$. It induces the joint distribution $P_{S_{1:K}, W_{1:K}, \overline{W}} = P_{\overline{W}|W_{1:K}} \prod_{i \in [K]} P_{S_i, W_i}$, where $P_{S_i} = \mu^{\otimes n}$ and $P_{\overline{W}|W_{1:K}} = \mathbb{1}_{\{\overline{W} = (W_1 + \cdots + W_K)/K\}}$. Similar to [YDVP20, BDP22], the population and empirical risks are defined as

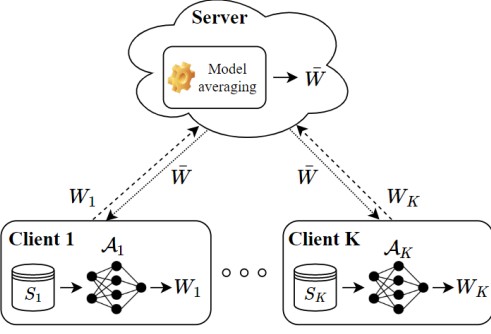

Figure 1: The considered distributed setup.

$$\mathcal{L}(\overline{w}) := \mathbb{E}_{Z \sim \mu}[\ell(Z, \overline{w})], \quad \hat{\mathcal{L}}(s_{1:K}, \overline{w}) := \frac{1}{K} \sum_{i \in [K]} \hat{\mathcal{L}}(s_i, \overline{w}). \qquad (2)$$

The main goal of the paper is to upper bound the generalization error $\text{gen}(s_{1:K}, \overline{w}) := \mathcal{L}(\overline{w}) - \hat{\mathcal{L}}(s_{1:K}, \overline{w})$ for the described (one-round) distributed learning setting as well as multi-round extension of it, i.e., Federated Learning. Results for the latter are stated in Section 5.

## 3 Information-theoretic bounds on the generalization error

In this section, we establish information-theoretic bounds on the generalization error of any stochastic distributed learning algorithm defined as in the previous section. It is important to note that the

bounds hold for $\overline{W}$ being any stochastic function of $W_{1:K}$. For the ease of the exposition, however, we only focus on the deterministic average case. The bounds are applied later for the DSVM and federated SGLD, suggesting a decreasing generalization error behavior as the number of clients increases.

## 3.1 Tail bound

A trivial approach that was already considered and used in [YDVP20] to establish in-expectation bounds for the problem studied therein consists in thinking of a distributed learning algorithm that is composed of $K$ local algorithms and an aggregator at the server as a single centralized algorithm. In other words, to consider the end-to-end system from the input at each local algorithm to the final aggregated model $\overline{W}$ as $P_{\overline{W}|S_{1:K}}$. The idea also applies to tail bounds. Hence, known tail and in-expectation bounds on the generalization error of centralized learning algorithms translate trivially into (generally loose) counter-part bounds for the distributed learning setting. Accordingly, the next theorem follows easily using the result of the above Theorem 1.

**Theorem 2.** *Suppose that the distributed learning algorithm $\mathcal{A}_{1:K}(S_{1:K})$ induces $P_{S_{1:K},\overline{W}}$ and suppose that for all $\hat{\overline{w}} \in \hat{\mathcal{W}}$, $\hat{\ell}(Z, \hat{\overline{w}})$ is $\sigma$-subgaussian. Then, for any fixed $\epsilon \in \mathbb{R}$ and $\delta \geqslant 0$, with probability at least $1-\delta$, $\mathrm{gen}\big(S_{1:K}, \overline{W}\big) \leqslant \sqrt{2\sigma^2(R_p(\delta,\epsilon) + \log(1/\delta))/(nK)} + \epsilon$, where $R_p(\delta,\epsilon) \coloneqq \sup_{Q:\, D_{KL}(Q\|P_{S_{1:K},\overline{W}})\leqslant \log(1/\delta)} \mathfrak{RD}(Q,\epsilon)$ and the supremum is over all possible distributions over $\mathcal{W} \times \prod_{i\in[K]} \mathcal{S}_i$.*

In the distributed setup, for every $i \in [K]$ only the hypothesis $W_i$ depends on the dataset $S_i$, which has then the effect with a factor of $1/K$ on $\overline{W}$. The result of Theorem 2, however, does not explicitly take the structure of the distributed learning problem into account and, instead, it considers the joint compressibility of all local algorithms and $P_{\overline{W}|S_{1:K}}$. Thus, the dependency of the bound on the structure of the problem is considered only implicitly, via the conditional distribution $P_{\overline{W}|S_{1:K}}$. In this work, we establish an alternate bound that is tailored specifically for the considered distributed setup. First, for every $i \in [K]$ we let the *compressed* hypothesis $\hat{\overline{W}}_i$ of $\overline{W}$ to depend on both $S_i$ and $W_{1:K\setminus i}$ (recall that $W_{1:K\setminus i}$ is independent from $S_i$). We then establish an upper bound on the generalization error in terms of the maximum (over all clients $i \in [K]$) of the minimum achievable compressibility using $P_{\hat{\overline{W}}_i|S_i,W_{1:K\setminus i}}$. The advantage can be exemplified as follows. For $i \in [K]$, consider only compressing the local algorithm $P_{W_i|S_i}$ and pick some $\hat{W}_i$ while keeping $W_{1:K\setminus i}$ un-compressed. Let $\hat{\overline{W}}_i$ be average of $\hat{W}_i$ and $W_{1:K\setminus i}$. As $\hat{W}_i$ has the effect with factor $1/K$ on $\hat{\overline{W}}$, in order to meet the distortion constraint $\mathbb{E}\big[\mathrm{gen}(S_i, \overline{W}) - \mathrm{gen}(S_i, \hat{\overline{W}}_i)\big] \leqslant \epsilon$ a more relaxed distortion criterion would be needed in compressing the local algorithm $P_{W_i|S_i}$. More precisely, for example when the loss is $\mathfrak{L}$-Lipschitz with $L_2^2$-norm, *i.e.*, $|\ell(z,w) - \ell(z,w')| \leqslant \mathfrak{L}\|w - w'\|^2$, a local distortion of $K^2\epsilon$ results into only $\epsilon$ distortion in the aggregated model.[1] Hence, this translates into smaller rate-distortion terms and, in turn, tighter bounds on the generalization error.

Now, to formally state the result, we introduce some definitions. Let $\mathrm{gen}(s_i, \overline{w}) \coloneqq \mathcal{L}(\overline{w}) - \hat{\mathcal{L}}(s_i, \overline{w})$. Besides, as in the centralized case, let $\hat{\mathcal{W}}$ be the alphabet of the compressed hypothesis and $\hat{\ell}: \mathcal{Z} \times \hat{\mathcal{W}} \mapsto \mathbb{R}^+$ a loss function (possibly different from $\ell$). Accordingly, for $\hat{\overline{w}} \in \hat{\mathcal{W}}$ and $s \in \mathcal{Z}^n$, let $\mathrm{gen}(s, \hat{\overline{w}})$ and $\mathrm{gen}(s_i, \hat{\overline{w}})$ be defined similarly with respect to $\hat{\ell}$. For a distribution $Q$ defined over $\mathcal{W} \times \prod_{i\in[K]}(\mathcal{S}_i \times \mathcal{W}_i)$, let

$$\mathfrak{RD}_i(Q,\epsilon) \coloneqq \inf_{P_{\hat{\overline{W}}_i|S_i,W_{1:K\setminus i}}} I(S_i; \hat{\overline{W}}_i | W_{1:K\setminus i}), \tag{3}$$

$$\text{s.t.} \quad \mathbb{E}\Big[\mathrm{gen}\big(S_i, \overline{W}\big) - \mathrm{gen}\big(S_i, \hat{\overline{W}}_i\big)\Big] \leqslant \epsilon, \tag{4}$$

where the infimum is taken over all Markov kernels (conditional distributions) of a random variable $\hat{\overline{W}}_i \in \hat{\mathcal{W}}$ given $(S_i, W_{1:K\setminus i})$. Note that the mutual information and expectations are with respect to

---

[1]For non-Lipschitz losses, such as 0-1 loss, the analysis is less trivial. An example can be found in Section D.2.

$Q_{S_i,,W_{1:K\setminus i}}P_{\widehat{\widetilde{W}}_i|S_i,W_{1:K\setminus i}}$ and $QP_{\widehat{\widetilde{W}}_i|S_i,W_{1:K\setminus i}}$, respectively, where $Q_{S_i,W_{1:K\setminus i}}$ is the $Q$-marginal of $(S_i, W_{1:K\setminus i})$.

**Theorem 3.** *Suppose that the distributed learning algorithm $\mathcal{A}_{1:K}(S_{1:K})$ induces $P_{S_{1:K},W_{1:K},\overline{W}}$ and suppose that for all $\widehat{\widetilde{w}} \in \hat{\mathcal{W}}$, $\hat{\ell}(Z,\widehat{\widetilde{w}})$ is $\sigma$-subgaussian. Then, for any fixed $\epsilon \in \mathbb{R}$ and $\delta \geqslant 0$, with probability at least $1 - \delta$,*

$$\mathrm{gen}\big(S_{1:K}, \overline{W}\big) \leqslant \sqrt{2\sigma^2\big(\max_{i\in[K]} R_p(\delta,\epsilon,i) + \log(1/\delta)\big)/n} + \epsilon, \tag{5}$$

$$R_p(\delta,\epsilon,i) \coloneqq \sup_{Q\colon D_{KL}(Q\|P_{S_{1:K},W_{1:K}})\leqslant\log(1/\delta)} \mathfrak{RD}_i(Q,\epsilon), \tag{6}$$

*where the supremum is over all possible distributions over $\prod_{i\in[K]}(\mathcal{S}_i \times \mathcal{W}_i)$.*

The theorem is proved in Appendix D.1. In the binary classification setup, the result also holds for the margin generalization error $\mathrm{gen}_\theta(S_{1:K},\overline{W})$ by letting $\mathrm{gen}_\theta\big(S_i,\overline{W}\big) \coloneqq \mathcal{L}(\overline{W}) - \hat{\mathcal{L}}_\theta(S_i,\overline{W})$.

While the advantage of this result was already stated in part above, it should be noted that the denominator in the result of Theorem 3 is $n$, rather than $nK$ in Theorem 2. In fact, none of the two results of Theorem 2 and Theorem 3 outperforms the other in general. For the case of distributed SVM which will be considered in the next section, it can be shown that the result of Theorem 3 results in a bound that decays with $K$ faster than one that uses Theorem 2. Moreover, as it will become clearer from the sequel the bounds for DSVM require a strict lossy compression, i.e., $\epsilon \neq 0$, and do not seem to be obtainable with a lossless compression framework, illustrating the utility of our rate-distortion based approach in general.

**Remark 3.** *For the ease of the presentation, the results of this work, including Theorem 3, are stated for the homogeneous case where the underlying data distribution $\mu_i$ is same for all clients. However, the result can be extended straightforwardly to the heterogeneous case, by considering $S_i$ and $\mathfrak{RD}_i(Q,\epsilon)$, $i \in [K]$, with respect to $\mu_i$.*

### 3.2 In expectation bound

Similar to Theorems 2 and 3, we establish upper bounds on the expectation of the generalization error of any distributed stochastic learning algorithm.

**Theorem 4.** *Suppose that the distributed learning algorithm $\mathcal{A}_{1:K}(S_{1:K})$ induces $P_{S_{1:K},W_{1:K},\overline{W}}$ and suppose that for all $\widehat{\widetilde{w}} \in \hat{\mathcal{W}}$, $\hat{\ell}(Z,\widehat{\widetilde{w}})$ is $\sigma$-subgaussian. Then, for every fixed $\epsilon \in \mathbb{R}$ we have:*

$$\mathbb{E}\big[\mathrm{gen}\big(S_{1:K},\overline{W}\big)\big]$$

$$\leqslant \frac{1}{n}\min\left\{\frac{1}{K}\sum_{j\in[n]}\sum_{i\in[K]}\sqrt{2\sigma^2\mathfrak{RD}(P_{Z_{i,j},\overline{W}},\epsilon)} + \epsilon, \sum_{j\in[n]}\sqrt{2\sigma^2\max_{i\in[K]}\mathfrak{RD}_i(P_{Z_{i,j},W_{1:K},\overline{W}},\epsilon)} + \epsilon\right\} \tag{7}$$

$$\leqslant \frac{1}{\sqrt{n}}\min\left\{\sqrt{2\sigma^2\mathfrak{RD}(P_{S_{1:K},\overline{W}},\epsilon)/K} + \epsilon, \sqrt{2\sigma^2\max_{i\in[K]}\mathfrak{RD}_i(P_{S_i,W_{1:K},\overline{W}},\epsilon)} + \epsilon\right\}. \tag{8}$$

The first terms of the minimization in (7) and (8) follow easily by an application similar to in Theorem 2. In particular, setting $\epsilon = 0$ in the first term of the minimization in (7) one recovers the result of [YDVP20, Thoerem 2] under the assumed subgaussianity. The second terms of the minimization in (7) and (8) are derived in a manner that is essentially similar to in the proof of Theorem 3. The details are omitted for brevity.

In Appendix C.2, it is shown that under the assumed subgaussianity, the upper bounds in [BDP22, Theorems 4 & 5] also can be recovered from Theorem 4. The results of [BDP22] are derived using the stability approach and by applying the *leave-one-out expansion* lemma.

## 4 Distributed support vector machines

In this section, we establish upper bounds on the generalization error of *Support Vector Machines* (SVM) [Vap06, CV95] when applied in a distributed learning setting. The algorithm is called hereafter

as *Distributed* SVM (DSVM). SVMs are popular and widely used for binary classification problems. They can be combined with different kernels in a computationally efficient way [BGV92, GKL20]. An easy application of SVM consists in finding a hyperplane based on a training dataset that separates the data with the smallest possible average margin error. Formally, let $\mathcal{Z} = \mathcal{X} \times \mathcal{Y}$, where $\mathcal{X} \in \mathbb{R}^d$ and $\mathcal{Y} = \{-1, +1\}$. The hypotheses are vectors $w \in \mathbb{R}^d$, which represent hyperplanes. For the simplicity of the exposition, we only consider the case with zero bias. The hyperplane predicts a label of a data $x$ according to the sign of the inner product $\langle x, w \rangle$. The 0-1 loss and margin loss functions are thus defined as $\ell_0(z, w) := \mathbb{1}_{\{y\langle x,w\rangle < 0\}}$ and $\ell_\theta(z, w) := \mathbb{1}_{\{y\langle x,w\rangle < \theta\}}$, respectively. In the distributed learning setup, each client $i \in [K]$ has access to dataset $S_i$ and picks the vector $W_i$; and the server node computes the aggregated model as $\overline{W} := (W_1 + \ldots + W_K)/K$ [Car20]. In this section, we study the generalization gap, which we denote hereafter as $\mathrm{gen}_\theta(S_{1:K}, \overline{W})$, defined as the difference between the population risk $\mathcal{L}(\overline{w})$ calculated using $\ell_0$ and the margin empirical risk $\frac{1}{K} \sum_{i \in [K]} \hat{\mathcal{L}}_\theta(S_i, \overline{W})$ calculated using $\ell_\theta$. It should be noted that the results of [BDP22, Theorems 4 & 5] require the loss function to be either a Bregman divergence or Lipschitz of some order; and, as such, they are not applicable to the DVSM setup that we consider here. This is because the 0-1 loss function is neither a Bregman divergence, nor a Lipschitz loss.

**Theorem 5.** *Let* $d \in \mathbb{N}^+$ *and* $\mathbb{P}(\|X\| \leqslant B) = 1$ *for some* $B > 0$. *Consider DSVM with* $K$ *clients each using* any *arbitrary local learning algorithm such that* $\mathbb{P}(\|W_i\| \leqslant 1) = 1$, $i \in [K]$.

*i) For any* $\delta > 0$, *with probability at least* $1 - \delta$,

$$\mathrm{gen}_\theta\big(S_{1:K}, \overline{W}\big) \leqslant \mathcal{O}\left( \frac{1}{nK\sqrt{K}} + \sqrt{\frac{\left(\frac{B}{K\theta}\right)^2 \log(nK\sqrt{K}) \log\big(\max\big(\frac{K\theta}{B}, 2\big)\big) + \log(1/\delta)}{n}} \right).$$

*ii) Also,*

$$\mathbb{E}\big[\mathrm{gen}_\theta\big(S_{1:K}, \overline{W}\big)\big] \leqslant \mathcal{O}\left( \frac{1}{nK\sqrt{K}} + \sqrt{\frac{B^2 \log(nK\sqrt{K}) \log\big(\max\big(\frac{K\theta}{B}, 2\big)\big)}{nK^2\theta^2}} \right).$$

This theorem is proved in Appendix D.2, by bounding the corresponding rate-distortion terms in Theorem 3 (for the part i.) and in the second term of (8) (for part ii.). To establish such bounds, we show the existence of a proper $P_{\hat{W}_i | S_i, W_{1:K \setminus i}}$ using techniques and results developed in [GKL20], and in particular by making use of the Johnson-Lindenstrauss transformation [JL84].

The above bound on the expectation of the generalization error of the DSVM decreases with $K$ with a rate of $\log(K)/K$. Moreover, it is important to note that the in-expectation bound for $K$ clients each having $n$ data samples is smaller than that of the counterpart centralized learning algorithm that has $nK$ input data samples by a factor of order $\mathcal{O}(\sqrt{\log(K)/K})$. This also holds for the tail bound, as long as $\log(1/\delta)/n$ is not the dominant term in the square root. Note that in general $\sqrt{\log(1/\delta)/n}$ is very small, corresponding to the generalization error of an dataset-independent algorithm [XR17].

**Remark 4.** *The tail bound in Theorem 5 for* $K = 1$ *does not recover the best known upper bound to the margin generalization error of SVMs. More precisely, for a centralized setup with dataset of size* $nK$, *[GKL20, Theorem 2] states*

$$\mathrm{gen}_\theta(S, W) \leqslant \mathcal{O}\left( \frac{B^2 \log(nK)}{nK\theta^2} + \sqrt{\frac{\left(\frac{B}{\theta}\right)^2 \log(nK) + \log(1/\delta)}{nK} \hat{\mathcal{L}}_\theta(S, W)} \right). \qquad (9)$$

*This is particularly important for the separable training dataset, where* $\hat{\mathcal{L}}_\theta(s, w) = 0$, *which makes the bound of order* $\mathcal{O}\big(B^2 \ln(nK)/(nK\theta^2)\big)$. *For the non-separable case, this term is asymptotically lower bounded by* $\hat{\mathcal{L}}_\theta(W^*)$, *where* $W^*$ *is the optimal population risk minimizer when* $\mu$ *is known. However, in the distributed learning setup, even if* $\hat{\mathcal{L}}_\theta(s_i, w_i) = 0$, $\hat{\mathcal{L}}_\theta(s_i, \overline{w})$ *is not necessarily zero.*

**Remark 5.** *The proof of Theorem 5 shows how our generalization bounds exploit the particular topology of the distributed learning setup. In particular, we show that for a fixed non-zero distortion level* $\epsilon$ *in (4) an upper bound on the rates in (4) scales as* $\mathcal{O}\big((\log(K)/K)^2\big)$. *When* $\epsilon = 0$, *a case for which the rate-distortion approach reduces to the mutual information-based approach (see the discussion right after Theorem 4), that upper bound scales only as* $\mathcal{O}(1)$. *In part this explains the benefits brought up by the rate-distortion approach upon the mutual information-based one.*

# 5    Federated stochastic gradient Langevin dynamics

In this section, we consider a *homogeneous federated learning* setup, denoted as FSGLD, where local learning algorithms use *stochastic gradient Langevin dynamics* (SGLD) method [WHGC21]. For the case of the centralized learning algorithm, [WHGC21] has proposed a new tractable upper bound on the generalization error, that describes well the generalization behavior, by using results of [BZV20] and by connecting SGLD to the Gaussian channels.

We consider a multiple-round distributed learning algorithm. At beginning of each round $t \in [T]$, the central server sends the updated hypothesis at the end of the previous round $\overline{W}_{t-1} \in \mathbb{R}^d$ to the clients, where $\overline{W}_0$ is a randomly initialized hypothesis. Then, each client $i \in [K]$ performs locally one iteration of SGLD, as explained later, and outputs $W_{i,t}$. The central server upon receiving these hypotheses choices, let $\overline{W}_t := (W_{1,t} + \cdots + W_{K,t})/K$. The final hypothesis $\overline{W}$ is chosen as a deterministic function of $\{\overline{W}_t\}_{t \in [T]}$. An example of such choices is Polyak averaging [PJ92], where $\overline{W} := (\overline{W}_1 + \cdots + \overline{W}_T)/T$.

Now, we explain the local SGLD algorithm applied by each client [GM91, WT11, WHGC21]. Each client $i \in [K]$, partitions its dataset $S_i$ into $m$ disjoint mini-batches $\{S_{i,1}, \ldots, S_{i,m}\}$, each one having equal size $b$ with elements $S_{i,j} = \{Z_{i,j,1}, \ldots, Z_{i,j,b}\}$. Then, client $i$ at round $t$, by receiving $\overline{W}_{t-1}$ from the central node, let $W_{i,t}$ be

$$W_{i,t} = \overline{W}_{t-1} - \eta_t \nabla_w \hat{\ell}(S_{i,j_t}, \overline{W}_{t-1}) + \sqrt{\frac{2\eta_t}{\beta_t}} V.$$

Here, $\eta_t$ is the learning rate, $\beta_t$ the inverse temperature, $V$ a $d$-dimensional random variable with distribution $\mathcal{N}(0, \mathrm{I}_d)$, where $\mathrm{I}_d$ is the $d \times d$ identity matrix, $j_t \in [m]$ is the mini-batch index, $\hat{\ell} \colon \mathcal{Z} \times \mathcal{W} \mapsto \mathbb{R}^+$ is a surrogate loss function, and

$$\nabla_w \hat{\ell}(S_{i,j_t}, \overline{W}_{t-1}) := \frac{1}{b} \sum_{l \in [b]} \nabla_w \hat{\ell}(Z_{i,j_t,l}, \overline{W}_{t-1}).$$

In the following we use the first term of (7) with $\epsilon = 0$ (which is reduced to the upper bound in [YDVP20, Thoerem 2]), to derive a bound on the expectation of the generalization error of FSGLD. Note that the first term of (7) is established by viewing the whole learning algorithm as a black-box, and hence it applies for the multiple-round federated learning setup as well, by considering $S_{1:K}$ as the training dataset and $\overline{W}$ as the chosen hypothesis.

**Theorem 6.** *Suppose that for each $w \in \mathcal{W}$, the loss $\ell(Z, w)$ is $\sigma$-subgaussian. The expected generalization error of FSGLD is upper bounded by*

$$\mathbb{E}\big[\mathrm{gen}\big(S_{1:K}, \overline{W}\big)\big] \leqslant \frac{\sqrt{2b}\sigma}{2nK\sqrt{K}} \sum_{j \in [m]} \sum_{i \in [K]} \sqrt{\sum_{t \in \mathcal{T}_{i,j}} \beta_t \eta_t \mathit{Var}\Big(\nabla_w \hat{\ell}(S_{i,j}, \overline{W}_{t-1})\Big)},$$

*where the set $\mathcal{T}_{i,j}$ contains the indices $t$ such that $j_t = j$ at client $i$, and*

$$\mathit{Var}\Big(\nabla_w \hat{\ell}(S_{i,j}, \overline{W}_{t-1})\Big) := \mathbb{E}\bigg[\Big\|\nabla_w \hat{\ell}(S_{i,j}, \overline{W}_{t-1}) - e_i\Big\|^2\bigg],$$

*where $e_i := \mathbb{E}\Big[\nabla_w \hat{\ell}(S_{i,j}, \overline{W}_{t-1})\Big]$.*

This result for the case of $K = 1$ is reduced in [WHGC21, Theorem 1] and it can be proved for any $K \in \mathbb{N}$ along the same lines of the proof of [WHGC21, Theorem 1]. Indeed,

$$\overline{W}_t = \overline{W}_{t-1} - \frac{\eta_t}{K} \left( \sum_{i \in [K]} \nabla_w \hat{\ell}(S_{i,j_t}, \overline{W}_{t-1}) \right) + \frac{1}{\sqrt{K}} \sqrt{\frac{2\eta_t}{\beta_t}} V', \tag{10}$$

where $V'$ is a $d$-dimensional random variable with distribution $\mathcal{N}(0, \mathrm{I}_d)$. Now, proceeding similar to the proof of [WHGC21, Theorem 1] concludes the result. The reason behind the extra factor $1/\sqrt{K}$ in the bound is that, when $K$ independent Gaussian noises are added, their variances are added up linearly with $K$. Thus, while in (10), the gradient is divided by $K$, the noise term is divided by $\sqrt{K}$,

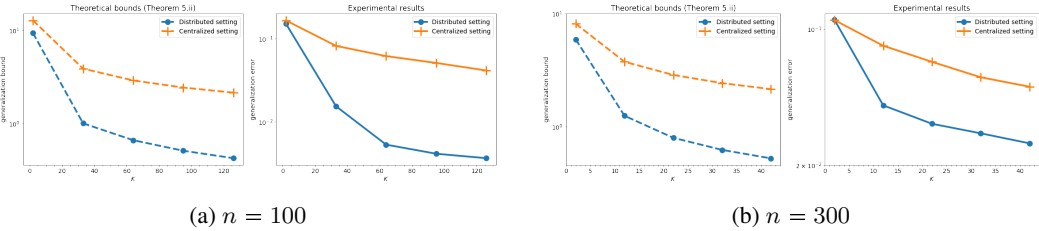

(a) $n = 100$                                    (b) $n = 300$

Figure 2: The generalization error for distributed and centralized SVM

which results in an extra $\sqrt{K}$ term in the denominator of the upper bound. The proof is omitted for brevity.

The established bound is decreasing with $K$, as far as the variance of the gradient with training data size $nK$ is not smaller than the variance of the gradient with training data size $n$, divided by $K$. Note that the variance of the gradient is correlated with the *sharpness* of the loss landscape and becomes small as the algorithm converges [WHGC21].

## 6  Experiments

This section aims at showing, through simulations, the evolution of the generalization error of the distributed SVM with the number $K$ of clients and the size $n$ of each individual training dataset. We will provide also comparison with the centralized setup, having training dataset of size $nk$, and also with the bounds on the generalization error as predicted by our analysis. We also provide further experiments and discussions on the comparison of the empirical and population risks of the distributed and centralized SVM. The details of the experiments are explained in Appendix B and here, we discuss our experimental findings.

**On the generalization error:**   Figure 2 shows the evolution of the generalization error (with zero margin) of the distributed setup ($K$ clients, each having $n$ data samples) as a function of the number $K$ of used clients for two values of $n$, $n = 100$ (Subfigure 2a) and $n = 300$ (Subfigure 2b). Also shown for comparison, the generalization performance of the associated centralized setup with a dataset of $nK$ samples.

The figure also depicts the evolution of the bound of our Theorem 5 (computed using part ii) of the theorem and with the value of the parameter $\theta$ set to $0.2$). For the centralized learning setting, the theoretical bound is obtained by applying the result of Theorem 5 (part ii) with the substitution $K_c = 1$ and $n_c = nK$.

Observe that, as predicted by our theoretical analysis of Section4, the system generalizes better in the distributed setup.

**On the empirical and population risks:**   While considering the empirical risk $\hat{\mathcal{L}}(s_{1:K}, \overline{w}) \coloneqq \frac{1}{K} \sum_{i \in [K]} \hat{\mathcal{L}}(s_i, \overline{w})$ is motivated by previous works [YDVP20, BDP22], it differs from the average local empirical risks minimized at clients, *i.e.,*

$$\tilde{\mathcal{L}}(s_{1:K}, w_{1:K}) \coloneqq \frac{1}{K} \sum_{i \in [K]} \hat{\mathcal{L}}(s_i, w_i). \tag{11}$$

Denote the difference of these empirical risks as

$$\Delta\hat{\mathcal{L}}(s_{1:K}, w_{1:K}, \overline{w}) \coloneqq \hat{\mathcal{L}}(s_{1:K}, \overline{w}) - \tilde{\mathcal{L}}(s_{1:K}, w_{1:K}). \tag{12}$$

Then, the population risk can be written as

$$\mathcal{L}(\overline{w}) = \tilde{\mathcal{L}}(s_{1:K}, w_{1:K}) + \text{gen}(s_{1:K}, \overline{w}) + \Delta\hat{\mathcal{L}}(s_{1:K}, w_{1:K}, \overline{w}). \tag{13}$$

The first term of the RHS of Eq. (13) can be made sufficiently small by minimizing the local empirical loss at each client. The second term is the generalization term considered in this paper, which is shown to decrease faster than the corresponding centralized case. In what follows, we study the evolution of the last term of the RHS of (13) with $K$.

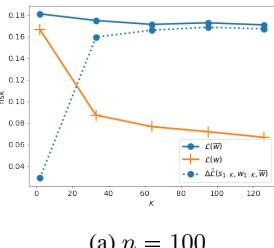

(a) $n = 100$

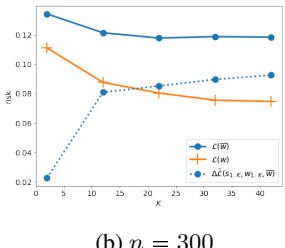

(b) $n = 300$

Figure 3: The population risk and empirical risk differences for distributed and centralized SVM

Indeed, as shown in Fig 3, while the considered generalization error decreases with $K$, the population risk of the centralized setup is smaller than the distributed setup. Also, the latter decreases more slowly with $K$. This is caused by the increase of term $\Delta\hat{\mathcal{L}}(S_{1:K}, W_{1:K}, \overline{W})$ with $K$ (note that this term is zero for the centralized setting). As it will become clearer from the rest of this section, in the distributed setup the term $\text{gen}(S_{1:K}, \overline{W})$ decreases with $K$ as specified by our Theorem 5 and the term $\Delta\hat{\mathcal{L}}(S_{1:K}, W_{1:K}, \overline{W})$ increases with $K$ (the term $\tilde{\mathcal{L}}(S_{1:K}, W_{1:K}) + \text{gen}(S_{1:K}, \overline{W})$ as defined by (11) can be made negligible for both centralized and distributed settings if local models are trained well enough, an assumption which will be made throughout hereafter). For the distributed case, the net effect is a decrease of the population risk with $K$, but which is slower than in the corresponding centralized setup. In what follows, we will show that as $K$ becomes large, if the clients use the same algorithm, i.e., $P_{W_i|S_i} = P_{W_1|S_1}$ for all $i \in [K]$, and they are trained well enough to minimize their respective empirical risks $\hat{\mathcal{L}}(S_i, W_i)$ then the population risk in the distributed case tends to a constant that depends only on $n$ and the used local algorithms, but not on $K$.

More formally, fix a training data size $n \in \mathbb{N}$ and suppose that all clients use the same local learning algorithm $P_{W_i|S_i} = P_{W_1|S_1}$. This empirical risks difference term is zero for $K = 1$, as $\overline{W} = W_1$. By increasing $K$, $\overline{W}$ departs further from $W_i$. In particular, as $K \to \infty$, $\overline{W} \to \mathbb{E}_{W_1}[W_1]$, where the expectation is with respect to the marginal distribution $P_{W_1}$ (induced by $P_{W_1|S_1}\mu^{\otimes n}$) and the convergence is almost surely due to the strong law of large numbers. Note the for a fixed data distribution $\mu$, the marginal distribution $P_{W_1}$ is a function of $n$ and local algorithms $P_{W_1|S_1}$. Hence,

$$\lim_{K \to \infty} \Delta\hat{\mathcal{L}}(S_{1:K}, W_{1:K}, \overline{W}) \stackrel{a.s}{=} \mathbb{E}_{S_1 \sim \mu^{\otimes n}}\left[\hat{\mathcal{L}}\left(S_1, \mathbb{E}_{W_1 \sim P_{W_1}}[W_1]\right)\right] - \mathbb{E}_{(S_1, W_1) \sim P_{S_1, W_1}}\left[\hat{\mathcal{L}}(S_1, W_1)\right]$$

$$= \mathbb{E}_{Z \sim \mu}\left[\ell\left(Z, \mathbb{E}_{W_1 \sim P_{W_1}}[W_1]\right)\right] - \mathbb{E}_{(S_1, W_1) \sim P_{S_1, W_1}}\left[\hat{\mathcal{L}}(S_1, W_1)\right]$$

$$= \mathcal{L}(E_{W_1}[W_1]) - \mathbb{E}_{(S_1, W_1) \sim P_{S_1, W_1}}\left[\hat{\mathcal{L}}(S_1, W_1)\right]. \tag{14}$$

The above is illustrated in Fig.3, where as visible from therein the population risk for the distributed setting (as computed experimentally) approaches the limit given by the RHS of (14) as $K$ gets large.

## 7 Concluding remarks

In this work, we established rate-distortion theoretic tail and in-expectation bounds on the generalization error of the distributed learning algorithms. Unlike previous approaches to this problem, our bounds, which are more general comparatively, are tailored specifically for the distributed setup. In particular, when applied to distributed SVM and FSGLD our results suggest a decreasing behavior of the generalization error with respect to the number of clients. The conducted experiments on DSVM are in accordance with the analytical results. Also, we partly investigated the evolution of the population risk with the number of clients and the size of the dataset. The analysis revealed the presence of a bias term. Possible directions for future works include (i) further investigation of the effect of the aforementioned bias term, (ii) study of the tightness of the proposed general bounds (experimentally), (iii) extension of the results to the setting in which the data distribution is not identical across the clients or when the training data samples are not independent, (iv) multiple-round scenarios, and (v) study of associated computational and communication costs.

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
