# OpenReview forum: "Rate-Distortion Theoretic Bounds on Generalization Error for Distributed Learning"
_NeurIPS.cc/2022/Conference — NeurIPS 2022 Accept_

### Official Review · Reviewer_HoVj · 2022-07-10

**Rating:** 6
**Confidence:** 4
**Soundness:** 3 good
**Presentation:** 3 good
**Contribution:** 2 fair

**Summary:**

This paper uses tools from rate-distortion theory to establish new generalization error upper bounds for distributed learning algorithms. The bounds depend on the compressibility of each client’s algorithm while keeping other clients’ algorithms un-compressed and leveraging the fact that small changes in each local model change the aggregated model by a factor of only $1/K$, where $K$ is the number of independent clients. The bounds are then applied to the distributed support vector machines (SVM), suggesting that the generalization error of the distributed setting decays faster than that of the centralized one with a factor of $O(\sqrt{\log(K)/K})$.

**Questions:**

My biggest concern in the paper is comparing the generalization error between distributed and centralized settings. It is stated after Theorem 5 that in expectation bound for $K$ clients, each having $n$ data samples is smaller than that of the counterpart centralized learning algorithm that has $nK$ input data samples by a factor of order $O(\sqrt{\log(K)/K})$. However, [BDP22, Theorem 4] states that the improvement is of order $O(1/\sqrt{K})$, which is even better. Furthermore, it is shown in Appendix B.3.2 that the upper bounds of [BDP22, Theorems 4 & 5] can be recovered as special cases and are subsumed by Theorem 4 in this paper. Thus, I believe that more explanation on the comparisons is needed.

Minor comments:
1.	In the abstract, it is said that “the generalization error of the distributed setting decays faster than that of the centralized one with a factor of $O(\log K/\sqrt{K})$.” I believe that it should be $O(\sqrt{\log(K)/K})$ as stated in Theorem 5.
2.	There is a missing reference in Remark 1 on Line 147.

---------------------
Post rebuttal:
Thanks for the clarification about the comparison with [BDP22, Theorem 4]. I did not notice the difference in the assumptions. It is a good paper, and I would like to maintain my original rating.




**Limitations:**

I think the authors addressed the limitations and potential negative societal impact of their work.



**Strengths And Weaknesses:**

This paper studies the generalization error of distributed learning algorithms using a novel tool of rate-distortion theoretic generalization bound developed by [SGRS22], which is sufficiently original. However, it is a bit unsatisfying to me that the results in this paper only tell us that the generalization error of a distributed algorithm is better than its centralized counterparts, but no more discussion on the computational and communication costs, and population risk. To my understanding, these aspects are the main reason to adopt distributed learning algorithms in practice instead of improving the generalization error.

The organization of the paper can be improved. I feel that some crucial contents are provided only in the Appendix, e.g., empirical validation, results of SGLD, and comparison with related prior art results. The proof details provided in Section 5 are not very informative (basically follows from [SGRS22] and [GKL 20]) and can be replaced by the additional results in Appendix.

---

> ### Author Response · Authors · 2022-08-02
> **Response to Reviewer HoVj**
>
> We would like to thank the reviewer for their interest and worthful comments. In the following, we address each remark/question with the most attention.
>
> **Concerns in the Weakness section:**
> * Regarding other aspects of distributed learning: We do agree with the reviewer. Analyzing the generalization error is only one of the steps to achieving a good understanding of the distributed learning algorithms. However, even for this aspect, there are very few works, and we believe our paper is one of the first general works that help to understand this problem in more depth. We have also some preliminary discussions about the population risk in the experiments section (that we will move to the main text), which need to be analyzed and continued in more detail in future works. We will make a more detailed discussion regarding this in the conclusion. And finally, we do share the opinion of the reviewer regarding the importance of the computational and communication aspects of distributed learning. Although they are not in the scope of this work, we should have mentioned these aspects in the introduction and the future directions. We will add these discussions as well.
> * Regarding the organization: We believe our proofs have considerable differences with \[SGRS22\] and \[GKL20\] while borrowing some crucial elements from these works. In particular, in proof of Theorem 3 intuitively we consider a double block coding technique (one more layer of block-coding technique than in \[SGRSS22\]), which is new. Moreover, while the proof of Theorem 5 uses the Johnson-Lindenstrauss transformation as in \[GKL20\], our proof technique is new and adapted to bound the rate-distortion terms (e.g. we don't have anymore the $\mathcal{E}_k$ events and the grid of vectors in $\mathbb{R}^k$ whose every entry is a whole multiple of $1/\sqrt{k}$, that are the main proof elements in \[GKL20, Theorem 2\]). More importantly, we thought this could help the reader to see how our generalization bounds make use of the particular topology of the distributed setup.
> However, we agree with the reviewer that having all the main results and experiments in the main text is a more reasonable choice. We will apply this change. Moreover, we will add a paragraph explaining the intuition (in particular for Theorem 5) behind how our generalization bounds make use of the topology of the distributed setup.
>
> **Concerns in the Questions section:**
> * Regarding the major concern: The result of \[BDP22, Theorem 4\] requires the loss function to be a Bregman divergence i.e. $\ell(w, z)=D_F(w, z)$ or $\ell(w,(x,y))=D_F(w^Tx,y)$, for some continuously differentiable and strictly convex function $F: \mathbb R^d \rightarrow \mathbb R$. In contrast, our general results (Theorems 2-4) do not require this, and instead only require a subgaussianity assumption (this assumption might be relaxed using a similar approach as in \[BZV20\] or by bounding the moment generating function in \[SGRS, Theorem 11\] with some other assumptions).
> In particular, DVSM, for which we specialized the generalized result and derived the additional factor of order $\mathcal{O}(\sqrt{\log⁡(K)/K})$ for the distributed setup in Theorem 5, is a binary classification problem. For this binary classification setup, as is usual, we considered the 0-1 loss function which is not a Bregman divergence, and hence \[BDP22, Theorem 4\] does not hold for this case. Indeed, the Bregman divergence $D_F(w, z)$ (similarly for $D_F(w^Tx,y)$) is defined to be convex in its first argument while the 0-1 loss function is not convex.
> Similarly, \[BDP22, Theorem 5\] is valid for a Lipschitz loss and when the variance of the hypothesis is bounded, and in particular, the 0-1 loss function is not a Lipschitz loss. Hence \[BDP22, Theorem 5\] also does not hold for this case.
> We add these discussions for more clarity.
>
> * Thanks for the minor comments; they will be modified.

---

### Official Review · Reviewer_7D8T · 2022-07-11

**Rating:** 6
**Confidence:** 3
**Soundness:** 3 good
**Presentation:** 3 good
**Contribution:** 2 fair

**Summary:**

The paper examines the generalization error of distributed learning using rate-distortion theory and compares the theoretical results with existing information-theoretical generalization bounds. For implication, the theory justifies the advantage of distributed learning by showing that the generalization error of the DSVM decreases with the client number K and is smaller than that of the counterpart centralized learning algorithm.

**Questions:**

- What if the sub-Gaussian assumption is removed?
	- Some work [1] shows that the parameters' distribution is heavy-tailed in multiple settings.
- What is the advantage of rate-distortion bounds over other generalization bounds (e.g., bounds via algorithmic stability or PAC-Bayes theory), especially in the distributed setting?
- How tight are the upper bounds derived in the paper? How big is $R_p$ in theorem 2 and theorem 3? Would $R_p$ be prohibitively large for deep complex models?
- In real-world settings, a fundamental challenge is that data may not be i.i.d.  across the workers. However, the generalization theory of distributed learning algorithms is mainly based on the assumption that the data is i.i.d. over clients. Can the results in this paper be extended to **non-i.i.d.** settings?

Reference

[1] Gurbuzbalaban, M., Simsekli, U. &amp; Zhu, L.. (2021). The Heavy-Tail Phenomenon in SGD. Proceedings of the 38th International Conference on Machine Learning, in Proceedings of Machine Learning Research.



**Limitations:**

No negative societal impact.

**Strengths And Weaknesses:**

**Strengths**
- The writing of the paper is good. The advantage of the technical approach adopted in this paper (i.e., the rate-distortion theory) is well discussed.
- As mentioned in the paper, previous results do not explicitly consider the structure of the distributed learning problem. The generalization bound derived in this paper is explicitly tailored for the considered distributed setting rather than a direct extension of the previous rate-distortion theoretic generalization bounds.
- The theory implies exciting and novel results: The generalization error of DSVM decreases with the client number K and is smaller than that of the counterpart centralized learning algorithm.

No major concerns.
Minor issue
- Line 264-265 seem to be missing something.

---

> ### Author Response · Authors · 2022-08-02
> **Response to Reviewer 7D8T (Part 1/2)**
>
> We thank the reviewer for their interest and their profound questions, which will be discussed below. The minor concern will be addressed directly.
> * Regarding subgaussianity: First, we recall that the subgaussianity assumption is for $\ell(Z,w)$ (similarly for $\ell'(Z,\hat{w})$) with respect to $\mu$, for any $w \in \mathcal{W}$. Hence, even if the parameters' distribution is heavy-tailed (as in \[1\]), this assumption could hold, e.g. for the 0-1 loss function.
> To answer the possible solutions without the subgaussianity assumption, we start by explaining its role in our results. For simplicity, consider Theorem 1, which recalls \[SGRS, Theorem 9\]. This result is derived in \[SGRS\] from \[SGRS, Theorem 11\] and by properly bounding the moment generating function (MGF) of a zero-mean r.v. using the Hoeffding inequality that holds due to the subgaussianity assumption. Without subgaussianity, this MGF should be bounded in a different way, probably with different assumptions. Some possibilities are using the cumulant generating function and the inverse of its Legendre dual, similar to \[BZV20\], or using Bennett's inequality (requires condition on the variance). These approaches, however, will introduce extra complexity in the presentation of our results; while for the considered cases as DVSM with 0-1 loss, the subgaussianity assumption holds. We will discuss these further potential extensions of our results.
> * Regarding RD approach: We found RD approach appropriate for establishing topology-dependant bounds on the generalization, suggesting a faster decay in comparison with the centralized setup. As it can be seen (paticularly in the proof of Theorem 5), this dependence manifests in our results mainly in the distortion criterion, i.e. Eq. (3). To explain this, as a valid choice, let $P_{ \hat{\overline{W_i}} |S_i,W_{1:K \setminus i}}=P_{\overline{W}|S_i,W_{1:K\setminus i}}$. Then, (3) holds for $\epsilon=0$ and the second term of (7) in Theorem 4 reduces to (upper bounded by) a mutual-information based (MI) bound $\mathbb{E}\left[\text{gen}(S_{1:K},\overline{W})\right] \leq \sqrt{2\sigma^2 \max_{i\in[K]} I(S_i;\overline{W}|W_{1:K\setminus i})/n}$, which is a worse bound than $\sqrt{2\sigma^2 \max_{i\in[K]} I(S_i;\overline{W})/n}$ (obtainable using the second term of (6) with $\epsilon=0$), due to the independence of $S_i$ and $W_{1:K\setminus i}$. Hence, for the zero distortion case, the bound reduces to an MI bound and as stated in \[BDP22\] for the similar results of \[YDVP20\], its dependence on the distributed setup is buried in the MI terms. This makes it difficult to compare the generalization bounds of the distributed and centralized setups, in general. However, it should be noted that for certain cases it is possible, as shown in the appendix for SGLD.
> PAC-Bayes bound are shown to have a close relation with MI-based bounds (e.g. Section 6.5.2. of Alquier, "User-friendly introduction to PAC-Bayes bounds"), and similarly, proposing proper topology-based bounds using this approach seems not to be straightforward. We emphasize that we do not claim achieving such bounds using these approaches is impossible. Rather, it seems that the dependence of the bound on the topology manifests more suitably in the distortion terms.
> In contrast, the algorithmic stability approach could be potentially a good candidate, as used in \[BDP22\] (by applying the Leave-one-out lemma). In the appendices, we showed that their results (with further subgaussianity assumption) are subsumed by our general results. Hence, at least in some cases, the RD approach includes also the algorithmic stability approach.
> We will provide a discussion about the intuition behind choosing the RD approach.

---

> > ### Author Response · Authors · 2022-08-02
> > **Response to Reviewer 7D8T (Part 2/2)**
> >
> > * Regarding tighntess of the bounds: This is a very fundamental question that concerns many of the previous works even for the centralized setup. To answer this, one needs to either characterize the generalization error (which seems to be infeasible except for toy examples), or establish a proper lower bound (which is notoriously a difficult problem), or verify the gap using experimental simulations. Our experiments for DSVM suggest that the bound in Theorem 5 captures the behavior of the generalization error, while order-wise not being tight. However, it should be noted that Theorem 5 is derived by **upper bounding** the rate-distortion terms in Theorem 3, hence Theorem 3 might give a tighter result. The exact estimation of these terms for large deep complex models seems to be out of reach at the moment. Further works are needed even for the centralized case to be able to assess these RD-based bounds (and MI-based bounds). We will discuss these aspects in the paper that could be interesting potential future directions.
> > * Regarding non-i.i.d. setup: We thank the reviewer for bringing up this discussion. We agree that in the real-world setting, the data is not necessarily i.i.d. across the workers. We distinguish two cases: the non-identical distribution across the clients and non-independent samples. The first case implies a setup in which it is assumed that the data of each user has a different distribution $\mu_i$, and the training dataset $S_i \sim \mu_i^{\otimes n}$, i.e. independent samples from $\mu_i$. Our framework can handle this setup as well. We only stated the result for the homogenous case merely for ease of presentation. We will add further discussions regarding the heterogeneous case. However, if the training samples are not independent, establishing generalization error even for the centralized case seems to be extremely difficult. This is due to the fact that most of the used concentration bounds work well for the independent samples. We discuss this limitation of our work, which is also the case for the previous works on the centralized setup.

---

### Official Review · Reviewer_8roe · 2022-07-12

**Rating:** 6
**Confidence:** 3
**Soundness:** 3 good
**Presentation:** 1 poor
**Contribution:** 4 excellent

**Summary:**

This is an information theoretical paper that considers distributed machine learning problems. New upper bounds on the generalization error of distributed learning algorithms are derived and applied to distributed support vector machines. In particular, a very surprising result is that the generalization error of the distributed SVM decays faster than that of the centralized SVM with a factor of O(log(K)/sqrt(K)), where K is the number of groups the whole data set is distributed.

======================

After studying the authors' response, I am happy to see that the background on rate-distortion theory is added, and the typos I found are taken care of. For these updates, I am happy to raise my score to 6.

**Questions:**

* Line 93, please clarify: is the output of Algorithm A random? That is, given exactly the same set of data and run A twice, would it be possible that A generates different output hypotheses (e.g., for the purpose of differential privacy etc.)? If not, how should we understand the conditional distribution P_{W|S} since it would then be a degenerated trivial distribution? If yes, would the author(s) provide more details on how the infimum between lines 116 and 117 is defined? Also, what is then the motivation of using a random algorithm in this paper?
* There seems to be one reference item missing in Line 147. Also, I think that the author(s) should declare that the distance d(w;w'|s) is not a distance in the mathematical sense since it lacks symmetry: d(w;w'|s) != d(w';w|s).
* Line 93, I guess that $\mathcal{W}\in\mathbb{R}^d$ should be $\mathcal{W}\subset\mathbb{R}^d$.
* Line 117: what is $\hat{\mathcal{W}}$?
* How is a Markov kernel defined?
* Line 138, is epsilon fixed or random?
* The assumption in Line 190 (sigma-subGaussian) is very difficult for me to follow. Is there any example of this (example that the assumption holds true, and also an example that the assumption is not true)?
* It seems that the function "gen(s,w)" not only depends on s and w, but also on the loss function, the distribution mu, and the sample size n. In particular, we see that in lines 117--118, the function "gen" depends on a new loss function $\hat{\ell}$ where it is emphasized that this may be different from $\ell$. Would it be possible that all the dependent elements be listed explicitly in different places where the function "gen" is used?
* Some long sentences need to be divided into shorter ones. For example the 4-line long sentence in Lines 107-110, the sentence in Lines 211-215, among others.
* In the definition of RD(Q,epsilon) between lines 116 and 117, what is the relation between $S$ and Q? What is the definition of $\hat{\mathcal{W}}$? How is Q deciding $\hat{W}$? What is the expectation taken with respect to? In particular, if the expectation is taken with respect to all the randomness, by definition E[gen(S,W)]=0, which is trivial. What is the role $W$ plays in the infimum? (See also my comments in the "Strengths And Weaknesses" section.)
* Equation (9), should [K] be [m]?
* how is the claim "the generalization error of the distributed setting decays faster than that of the centralized one with a factor of O(log(K)/sqrt(K))" supported by, e.g., Theorem 5?

**Limitations:**

There is no negative societal impact observed. The limitations focus on that we doubt if all the claimed results are well supported.

**Strengths And Weaknesses:**

Strengths:
* solid theoretical analysis
* the results on distributed SVM are very surprising

Weaknesses:
* one of the key tools used throughout the paper, the function RD, is not defined clearly enough (see comments below in the "Questions" section), rendering the whole paper very difficult to follow.
* From bounds on the generalization error of some machine learning algorithm, readers could expect to quickly find out how the generalization error converges to zero (or not), as the training sample size tends to infinity, or while other variables are changing. However, in this paper, such reading is not easy. For example, in Theorem 2, the right-hand side of the bound contains Rp(delta, epsilon), which is defined in a highly complex way such that even telling whether this quantity would be O(1) is extremely difficult. Theorems 3 and 4 are similar.
* I am not sure if the claimed results are well supported.

---

> ### Author Response · Authors · 2022-08-02
> **Response to Reviewer 8roe (Part 1/3)**
>
> We thank the reviewer for their careful reading of the paper and their comments. The concerns of the reviewer are mainly regarding the presentation of the paper, as they evaluated our contribution as Excellent (4/4). In the following discussions, and by adding a new section in the Appendix explaining the related concepts and definitions of the rate-distortion theory in the next version, we believe the concerns of the reviewer are addressed. Please do let us know if further clarifications are needed.
>
> **Concerns in the Weakness section:**
> * Regarding RD background: Thanks for the feedback. The rate-distortion theory and functions are very well known and are thoroughly explained in the lossy compression context in \[Ber75, CT06, PW14\] and in the learning algorithm context (as used in this paper) in \[Sections 2.2. & 3.1, SGRSS22\]. They measure the "lossy compressibility amount" of a learning algorithm which is known to be related to good generalization behavior for many years (e.g. \[Littlestone and Warmuth 1986\]). In the paragraph before Theorem 1, we briefly introduced the main ideas and concepts of RD used in our work and what "lossy compressibility" means. But based on your feedback, we agree that to make the paper self-contained, the related background and concepts could be expanded more. We will explain this in more detail, particularly by adding a new section in the Appendix, dedicated to RD concepts and definitions.
> * Regarding the order of bounds: We understand the concern of the reviewer. This is however the case for many works on the generalization error, e.g. all information-theoretic bounds initiated by \[RZ16, XR17\], the bounds in terms of variance of the gradient (e.g. \[WHGC21\]), etc. Besides, while a general upper bound on the RD term cannot be found, it can be computed or bounded in many cases:
>    * When $\mathcal{W}$ is a finite set, the RD function is bounded by $\log(|\mathcal{W}|)$.
>    * When $\mathcal{W} \subset \mathbb{R}^d$ is inside a $d$-dimensional ball of radius $R$ ($V_R$) and when the loss is $\mathfrak{L}$-Lipschitz with respect to $L_2$ distance, then $\mathfrak{RD}(Q,\epsilon)$ is upper bounded by $N_{R,\epsilon'}$, the $\epsilon'$-covering number of $V_R$, where $\epsilon'=\epsilon/(2\mathfrak{L})$.
>    * For many known distributions, the closed form solution of the RD function exists, see e.g. \[CT06\]. As an example, for a $d$-dimensional i.i.d. Gaussian vector $W \sim \mathcal{N}(0,\sigma^2\mathrm{I}_d)$, the RD function with respect to $L_2^2$ distance equals $\frac{d}{2}\log(\max(\sigma^2/\epsilon,1))$.
>    * In many cases, the RD function for small values of $\epsilon$ can be well approximated by Shannon "lower bound" (which becomes tight as $\epsilon \rightarrow 0$, \[T. Koch, 2016\]).
>    * Having the Lipschitz assumption, there exists many computationally efficient algorithms, e.g. Blahut-Arimoto algorithm, that can compute the RD function for discrete (and continuous by first making a fine quantization) alphabets.
>    * Tighter bounds on the RD term can be found depending on the setup, as done in the proof of Theorem 5 without dependency on the dimension $d$.
> We will add these discussions to the new section that will be added.
> * Regarding our results: Our theoretical results are rigorously proved and are quite general. They are also consistent with our experimental findings, reported in the Appendix (we will move them to the main text for more visibility). We agree that this would be interesting to verify our findings in other setups, for example for non-i.i.d.  datasets. We will discuss these aspects for future works in the conclusion.

---

> > ### Author Response · Authors · 2022-08-02
> > **Response to Reviewer 8roe (Part 2/3)**
> >
> > **Concerns in the Questions section:**
> > * The considered framework in this work is one of the most general frameworks in statistical learning, used in many previous works (e.g. \[XR17\]). In its most general case, we consider the learning algorithm to be potentially random. This is indeed the case for many practical optimization algorithms like SGD. As an example, the stochasticity of SGD comes from the random initialization of the hypothesis, random choices of the mini-batches from the dataset in each iteration, etc. Hence, if one fixes the dataset and all the related hyperparameters, the chosen output is with high probability different in every realization. In algorithms like SGLD, extra stochasticity also arises from the added gaussian noise in each iteration. Thus, overall the randomized algorithms are widely used in practice and this is the main motivation for considering this general framework.
> > Regarding the infimum, as explained below, it is taken over all possible Markov kernels $P_{\hat{ W }|S}$, which intuitively means over all possible "compressed (random) learning algorithms". Note that this paper deals with the upper bounds on the generalization error and hence for example in Theorem 1, any choice of such valid $P_{\hat{W}|S}$ results in an upper bound. Thus, one does not necessarily need to find the infimum (which indeed gives the best upper bound in Theorem 1).
> > * Thanks for the missing reference; will be added. We agree with the reviewer that using "distance" for $d$ is not mathematically correct. In the next version, we will change it to "distortion function".
> > * Thanks; will be modified to $\mathcal{W} \subset \mathbb{R}^d$.
> > * The set $\hat{\mathcal{W}}$ is the support set of the "compressed" (or quantized) hypothesis $\hat{W}$. This set can be any arbitrary set (for example in the proof of Theorem 5), but in many cases (e.g. when we consider quantization of the space) it is a subset of $\mathcal{W}$. We explain this set with some concrete examples in the new section of the Appendix.
> > * Once the set $\hat{\mathcal{W}}$ is fixed, the set of Markov kernels $P_{\hat{W}|S}$ is composed of all possible conditional distributions of random variables $\hat{W} \in \mathcal{\hat{W}}$ given $S$. This means that the only needed condition is having $\int_{\hat{w}\in \hat{\mathcal{W}}}  \mathrm{d}P_{\hat{W}|S=s}=1$ for all $s \in \mathcal{S}$. Note that $P_{\hat{W}|S}$ together with $P_S$ define the joint distribution of $(\hat{W},S)$. This set of Markov kernels (in the infimum) is not new and was introduced before both in the lossy source coding context \[PW14\] and in the learning algorithm context \[SGRS22\]. Further explanations about the Markov kernels will be added also in the new section in the Appendix.
> > * The theorem is valid for any fixed value of $\epsilon$. This will be made precise.
> > * The subgaussianity is  a very common assumption, appeared previously in many works (e.g. \[XR17\]). As an example, if for all $z,\hat{\overline{w}}$, we have $\hat{\ell}(z,\hat{\overline{w}}) \in [a,b]$ (the loss is bounded), then $\hat{\ell}(Z,\hat{\overline{w}})$ is $\frac{(b-a)}{2}$-subgaussian, due to Hoeffding's Lemma. Hence, this includes the 0-1 loss function. For an example of the case where this does not hold, consider $\hat{\ell}(Z,\hat{\overline{w}})$ having a fat-tailed (called also heavy-tailed) distribution, or more generally any distribution having an infinite variance, such as the Cauchy distribution. We will add the example of the bounded loss for clarity.
> > * As mentioned, the generalization error $\text{gen}(s,w)$ does depend on the learning algorithm ($P_{W|S}$), the loss function,$\mu$, and $n$ (as a summary it depends on $P_{S,W}$ and the loss function ). Although we believe that this is the classical definition and notation of the generalization error (which appeared in many previous works) and the mentioned dependencies are clear from the context, we will further emphasize this in the first paragraph of Section 2.
> > It should be noted that the generalization error $\text{gen}(S,W)$ does not depend on the choice of $\hat{\ell}$, but rather our proposed bound on $\text{gen}(S,W)$ depend on that. To establish such bound, intuitively instead of directly bounding $\text{gen}(S,W)$, we first bound $\text{gen}(S,\hat{W})$ (which depends on $\hat{\ell}$) and then we add $\epsilon$, the average difference of these two quantities.   In other words, in our results, $\hat{\ell}$ is used as an extra degree of freedom in the generalization bound, allowing potentially to have tighter bounds.

---

> > > ### Author Response · Authors · 2022-08-02
> > > **Response to Reviewer 8roe (Part 3/3)**
> > >
> > > * Thanks; long sentences will be rephrased.
> > > * Regarding the terms related to RD function definition:
> > >    * In Theorem 1 (and similarly in Theorem 3), the bound on the generalization error depends not only on the RD function with respect to $P_{S,W}$ but also with respect to all joint distributions $Q$ of $(S,W)$ that are close enough to $P_{S,W}$ (i.e. $D_{KL}(Q\|P_{S,W})\leq \log(1/\delta)$). For this reason, in the equation after line 116, we defined the RD function for any joint distribution $Q$ of $(S,W)$ (with marginals $Q_S$ and $Q_W$), that includes $P_{S,W}$ as a special case. The set $\hat{\mathcal{W}}$, explained in the above comments, can be chosen arbitrarily (as far as there exists a proper loss function $\hat{\ell}$ for that set) and does not depend on $Q$. The expectation term is with respect to $QP_{\hat{W}|S}$, as stated in line 118. These are the common notations/definitions in the rate-distortion theory \[Ber75,CT06,PW14,SGRS22\].
> > >    * The term $\mathbb{E}[\text{gen}(S,W)]$ is not equal to zero in general: It can be written as $\mathbb E_{(S',W) \sim P_S P_W}[\hat{\mathcal{L}}(S',W)] - \mathbb E_{(S,W) \sim P_{S,W} } [\hat{\mathcal{L}}(S,W)] $, which is not zero in general. See for example \[XR17\], for some bounds on this term.
> > >    * As it is clear from the equation, the role of $W$ in the infimum is hidden in the condition that $P_{\hat{W}|S}$ needs to satisfy.
> > > We believe adding a new section in the appendix and carefully explaining these definitions (accessible in \[Ber75,CT06,PW14,SGRS22\]) will resolve most of the concerns of the reviewer.
> > > * Thanks for catching this typo; in the first line, $[K]$ will be changed to $[m]$.
> > > * Consider for example the bound in Theorem 5.ii and its behavior w.r.t. $K$: for the centralized case (one client) with $nK$ training data samples ($K\mapsto 1,$$n \mapsto nK$), it is $\mathcal{O}\left(\sqrt{\frac{\log{K}}{K}}\right)$, and for the distributed case, with $K$ clients each having $n$ samples, it is $\mathcal{O}\left(\sqrt{\frac{\log(K\sqrt{K})\log(K)}{K^2}}\right)=\mathcal{O}\left(\frac{\log(K)}{K}\right)$, as $\mathcal{O}\left(\log(K\sqrt{K})\right)=\mathcal{O}\left(\log(K)\right)$. It should be noted that as we have emphasized in the paper, the established bounds on the generalization error "suggest" such behavior. This is further verified experimentally for the DSVM in the appendix.

---

### Author Response · Authors · 2022-08-02
**Thanks to all reviewers**

We thank the reviewers for the time and effort spent in reviewing the paper and for their thoughtful comments. We are glad to see that all reviewers overall found our results interesting. Below, we address the concerns of each reviewer separately. To avoid repeating the concerns/questions, we address them in the order of appearance in the Weakness and Questions sections.

---

### Author Response · Authors · 2022-08-06
**Thanks again to all reviewers**

We thank the reviewers once again for their valuable feedback. We have provided detailed discussions and responses to all comments and concerns of the reviewers separately. As the discussion period is ending in a couple of days, we would like to kindly ask the reviewers if all their concerns have been sufficiently addressed or if further clarification is needed.

 Best regards,
Authors

---

### Author Response · Authors · 2022-08-09
**The revised version**

Once again we thank the reviewers for their useful comments which helped improve the quality of our paper. We have uploaded a revised version of the paper in which we addressed all the concerns and suggestions of the reviewers. In particular, as suggested by Reviewer 8roe we have added a new 3-page length section in the supplements to recall the rate-distortion theoretic concepts and definitions. Moreover, as per the suggestion of Reviewer HoVj, we moved some results from the supplemental material to the main body of the text. If, hopefully, the paper is accepted, we will have an extra one page for the camera-ready version; and so we can move more results/experiments to the main text. Finally, we elaborated a bit more on the comparison with the prior art results of [BDP22, Theorems 4 and 5].

Best regards,
Authors

---

### Meta-Review · Area_Chair_64f3 · 2022-09-07

**Recommendation:** Accept
**Confidence:** Certain

**Metareview:**

The paper investigates generalization error bounds of distributed learning algorithms via rate-distortion theory, and compares the theoretical results with existing information-theoretical generalization bounds. An interesting implication of the authors' theory is the advantage of distributed learning with multiple clients as opposed to a centralized algorithm. This applies, in particular, to distributed SVM and distributed SGD, under some variance regimes. The paper strengthens the literature in several respects by removing or relaxing assumptions.
All in all, an interesting piece of work that has improved during the discussion period. The authors are committed to re-organized their submission according to what they proposed during the discussion.



**Award:**

No

---

### Decision · Program_Chairs · 2022-09-14

Accept